# Diffusion-Based, Data-Assimilation-Enabled Super-Resolution of Hub-height Winds

## Abstract

High-quality observations of hub-height winds are valuable but sparse in space and time. Simulations are widely available on regular grids but are generally biased and too coarse to inform wind-farm siting or to assess extreme-weather-related risks (e.g., gusts) at infrastructure scales. To fully utilize both data types for generating high-quality, high-resolution hub-height wind speeds (tens to 100m above ground), this study introduces `WindSR`, a diffusion model with data assimilation for super-resolution downscaling of hub-height winds. `WindSR` integrates sparse observational data with simulation fields during downscaling using state-of-the-art diffusion models. A dynamic-radius blending method is introduced to merge observations with simulations, providing conditioning for the diffusion process. Terrain information is incorporated during both training and inference to account for its role as a key driver of winds. Evaluated against convolutional-neural-network and generative-adversarial-network baselines, `WindSR` outperforms them in both downscaling efficiency and accuracy. Our data assimilation reduces `WindSR`'s model bias by approximately 20% relative to independent observations.

## 1 Introduction

Wind varies strongly in space and time and is shaped by land use/land cover, topography, and synoptic-scale circulation patterns Peco et al. (2025); Jung et al. (2025); Sheridan et al. (2023). Accurate characterization therefore requires fine spatial and temporal resolution. While thousands of in situ observations exist near the surface (e.g., 10 m; Automated Surface/Weather Observing Systems, ASOS), measurements at rotor heights (tens to 100 m) remain scarce. For example, Peco et al. (2025) compiled public and partially restricted towers data across North America ( including Alaska) and found only 25 locations with rotor-height winds, spanning 1-20 years and 13-100 m. These differences in observation periods and heights across sites make robust wind resource assessments and gust-risk evaluation challenging. As a result, the wind community and utilities often rely on numerical or statistical downscaling to estimate wind speeds at kilometer- to tens-of-kilometers-scales (e.g., Tang et al. (2016); Draxl et al. (2015; 2024)).

Two conventional downscaling approaches are used. Dynamical downscaling solves the governing equations to simulate atmospheric processes and produce comprehensive data, including wind, solar, and hydro resources, but it is computationally expensive for long periods and large domains. Statistical downscaling builds empirical links between coarse-resolution predictors and local-scale climate variables Schoof (2013). It is computationally efficient and targets variables of interest (e.g., wind speed) but depends heavily on observations, which are especially challenging for winds at hub-height levels. Recently, machine learning (ML) methods have combined statistical efficiency with physically informed features/constrains Sachindra et al. (2018); Jebeile et al. (2021); Yeganeh-Bakhtiary et al. (2022), yielding more physically consistent results (e.g., Wang et al. (2021)). However, most ML downscalers primarily add high-resolution detail to coarse-resolution climate model outputs without correcting biases; consequently, the downscaled fields inherit upstream errors from the original coarse-resolution models. This is an important limitation of current high-resolution wind datasets such as Sup3rCC Buster et al. (2023). While these datasets are highly valuable,

accuracy can be further improved by incorporating observational information available at sparse locations and times — e.g., through data assimilation (DA).

DA is a widely used technique in weather-forecast models, such as the U.S.-developed Global Forecast System, and the High-Resolution Rapid Refresh (HRRR). DA integrates observations from satellites, weather stations, and other sources (e.g., radar, sounding) to refine forecasts, effectively bridging the gap between numerical models and real-world measurements  Navon (2009). It is especially valuable for variables that are difficult to simulate accurately—such as wind and precipitation—because they vary sharply in space and time and depend on the accurate simulation of other atmospheric processes  Županski & Mesinger (1995); Cheng et al. (2017). Recently, DA has been incorporated into ML-based forecasting models, improving accuracy and speed across many applications. For example, Maulik et al. (2022) demonstrated that adding DA to a long short-term memory forecasting model improved 500-hPa geopotential-height predictions with an 18-day lead time compared with the same model without DA. However, geopotential height is a comparatively smooth variable with relatively small spatial variability, making it far less challenging to predict than highly variable fields like wind. Thus, applying DA and ML to more complex variables such as wind remains a critical challenge.

This study introduces `WindSR`, which couples data assimilation with diffusion-based super-resolution (SR) downscaling. `WindSR` employs Denoising Diffusion Probabilistic Models (DDPM) Ho et al. (2020) and conditions the diffusion process on a blended field that merge sparse observations with gridded simulations via a dynamic impact-radius scheme. This method provides spatially adaptive conditioning that mitigates simulation biases where observations exist. Terrain features are included as auxiliary conditioning during training and inference to enhance physical realism.

The main workflow consists of three key steps: (1) training a deep-learning SR model for windspeed with terrain information; (2) blending sparse but valuable observations into the numerically simulated winds using a dynamic-impact radius; and (3) conditioning the diffusion-based downscaling with this blended data.

We summarize our contributions as follows: (1) A DA-conditioned diffusion framework for wind SR (`WindSR`) with terrain-aware conditioning. (2) A dynamic impact-radius assimilation mechanism that adapts observation influence in space. (3) Comprehensive evaluation against convolutional-neural-network and generative-adversarial-network (CNN/GAN) baselines, demonstrating higher accuracy and efficiency, including  20% bias reduction relative to independent observations.

This paper is organized as follows: section 2 reviews related work, including recent diffusion-model applications in climate science. section 3 introduces preliminaries, section 4 describes our method, our experiments and results are presented in section 5, and we conclude our work in section 6.

## 2 Related Work

This section reviews relevant work and techniques aimed at our goal: developing ML models that generate high-resolution wind fields with fine detail for wind energy, while reducing biases in these models through the assimilation of observational data.

**Deep learning for super-resolution.** Deep learning has transformed image SR, which reconstructs high-resolution images from low-resolution inputs. A foundational deep learning model is the SR convolutional neural network (SRCNN) (Dong et al., 2015), which introduced an end-to-end approach with three stages: patch extraction and representation, non-linear mapping, and reconstruction. The SRCNN (Dong et al., 2015) demonstrated large gains over traditional methods such as bicubic interpolation, establishing a new standard for image quality.

Building on this progress, the Enhanced SR GAN (ESRGAN)  (Wang et al., 2018) uses a generative adversarial framework to produce sharper and more realistic high-resolution images. ESRGAN comprises a generator that creates high-resolution outputs and a discriminator

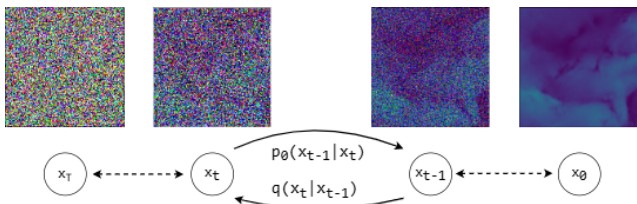

Figure 1: The figure demonstrates the DDPM's forward process of adding noise and the reverse denoising process.

that evaluates them against reference samples, driving iterative improvement. Stengel et al. 2020 Stengel et al. (2020) applied ESRGAN to downscale future wind and solar projections from 100 km to 2 km. Dettling et al. 2025 Dettling et al. (2025) trained an ESRGAN on large-eddy simulations (LES) to downscale from 960 m to 30 m over complex terrain in the northwestern U.S.; the model, tested in a different region, demonstrated credible performance in terms of energy spectra and flow statistics.

**Diffusion models for super-resolution.** Super-Resolution via Repeated Refinement (SR3, Saharia et al. (2021)) is a diffusion model tailored to image SR that leverages DDPM Ho et al. (2020). Starting from a noisy input, SR3 progressively denoises to recover high-resolution detail while conditioning on the low-resolution image to preserve consistency with the original image context. During training, SR3 learns the reverse of the noise-adding process; during inference, it applies the learned reverse process, starting from a low-resolution image, and gradually reconstructing a high-resolution image.

**Diffusion models for weather applications.** Diffusion models have seen increasing use for weather data, often delivering more accurate and efficient data generation than GAN frameworks (e.g., Chen et al., 2023). Their relevance arises in three ways. First, diffusion models originated in image generationHo et al. (2020). Weather fields at a given time step can be viewed as images whose patterns vary substantially between times. Unlike human faces or animals, winds lack prior spatial structure and exhibit sharp gradients, making them harder to learn, diffusion models handle such complexity well. For example, Kurinchi-Vendhan et al. (2024) (Kurinchi-Vendhan, 2024) used an SR3-based approach with quantile regression to generate high-resolution wind components within specified bounds, demonstrating greater accuracy of diffusion models for wind applications. Second, unlike deterministic downscalers, diffusion models naturally generate ensembles by sampling from learned distributions, enabling uncertainty and long-term trend quantification. Ling et al. (2024) (Ling et al., 2024) employed a diffusion-probabilistic downscaler to create a 180-year monthly surface dataset for East Asia, enhancing resolution of global climate data from 1° to 0.1° and yielding local-scale insights into long-term trends important for climate-adaptation and infrastructure planning. Finally, the stepwise Markov process and probabilistic framework of diffusion models offer a robust pathway to incorporate DA. Huang et al. (2024) Huang et al. (2024) demonstrated a denoising diffusion model that assimilates reanalysis and sparse observations, reporting 48-h forecasting with accuracy comparable to 24-h baselines.

## 3 Preliminary

### 3.1 Denoising Diffusion Probabilistic Models

Denoising Diffusion Probabilistic Models (DDPMs) Ho et al. (2020) are generative models that synthesize high-quality samples by learning to reverse a gradual noising process. These models consist of two main processes: a forward process and a reverse process, as shown in Figure 1. In the forward (diffusion) process, Gaussian noise is incrementally added to the data over a series of forward steps, ultimately transforming the original image into near-white noise after $T$ time steps. At each time step $t$, a small amount of noise is added according to a predefined schedule $\beta_t$. This schedule determines the rate and extent of noise added at each step.

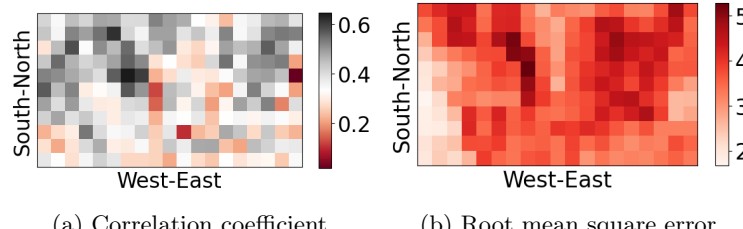

(a) Correlation coefficient  (b) Root mean square error

Figure 2: Correlation coefficient (left) and root mean square error (right) between WTK and HRRR hourly data averaged over 5 days over each 128x128 tile. Despite advances in NWP, significant model biases remain, especially at coarser resolutions and in areas where complex terrain makes parametrization difficult. Data driven approaches can rectify these differences without the need for computationally expensive physics.

$$q(\mathbf{x}_t \mid \mathbf{x}_{t-1}) = \mathcal{N}\left(\mathbf{x}_t; \sqrt{1 - \beta_t}\mathbf{x}_{t-1}, \beta_t \mathbf{I}\right) \tag{1}$$

In the reverse process, DDPMs undo the forward process step-by-step, starting from random noise and iteratively refining the sample to generate images. The reverse process is guided by a deep neural network denoiser (typically a U-Net), which is trained to predict the injected noise at each step in the forward process, and gradually learns to denoise the image.

$$p_\theta(\mathbf{x}_{t-1} \mid \mathbf{x}_t) = \mathcal{N}\left(\mathbf{x}_{t-1}; \mu_\theta(\mathbf{x}_t, t), \sigma_t^2 \mathbf{I}\right) \tag{2}$$

When doing training for the neural network model, the objective is to approximate the reverse distribution such that we can effectively denoise the noisy samples; and to minimizes the difference between the true noise $\epsilon$ and the predict noise $\epsilon_\theta$. The training loss can be expressed as:

$$L_t = \mathbb{E}_{\mathbf{x}_0, \epsilon, t}\left[\|\epsilon - \epsilon_\theta(\mathbf{x}_t, t)\|^2\right] \tag{3}$$

Where $\epsilon \sim \mathcal{N}(0, \mathbf{I})$ represents the Gaussian noise, and $\epsilon_\theta(\mathbf{x}_t, t)$ is the model's predicted noise.

### 3.2 DATA

This study presents a framework to produce high-fidelity, high-resolution wind fields at turbine-rotor height and wind-farm scales, with improved accuracy relative to existing datasets that omit DA. Ideally, two data types are required: (1) training data at dozens to hundreds of meter resolution, also known as Large Eddy Simulations (LES). However, at the time of development, no sufficiently long-term LES datasets were available; and (2) observational sites at a uniform height over at least a subregion of the U.S. As noted in Introduction, the only 25 hub-height sites span different heights (13 to 100m above ground level). To advance the framework despite these constraints, we trained on the 2 km WIND Toolkit (WTK) data and used randomly sampled High-Resolution Rapid Refresh (HRRR) fields as observational inputs for DA. Once validated, the framework can be re-trained for regions where suitable LES data exist.

**Data for diffusion SR** We use the latest WIND Toolkit (Draxl et al. (2024)), which provides wind speed and direction at 2km spatial and 5-minute temporal resolution for 2018–2020 at multiple heights(10-500 m above ground level). For diffusion SR training, windspeed at 80 m above ground level were calculated hourly for 2018-2019. Static terrain height was included to account for topographic influences Data from 2020 were reserved for testing. Further details appear in Table 1.

**Data for diffusion DA** The HRRR dataset, developed by National Oceanic and Atmospheric Administration (NOAA), serves as "ground truth" for DA. HRRR has 3 km spatial and hourly temporal resolution and assimilate diverse obserations (e.g., surface observations,

| Data | WTK | HRRR | HGT |
|------|-----|------|-----|
| Type | Wind | Wind | Terrain |
| Spatial | 2 km | 3 km | 2 km |
| Temporal | 5 min | 15 min | Constant |
| Years | 2020 | 2020 | N/A |
| HR | $128 \times 128$ | $128 \times 128$ | $128 \times 128$ |
| LR | $16 \times 16$ | $16 \times 16$ | $16 \times 16$ |

Table 1: Dataset specifications for WTK, HRRR, and HGT

radiosondes, satellites, radars, aircrafts Dowell et al. (2022); James et al. (2022). HRRR has been extensively validated for wind assessment across onshore (including complex terrain) and offshore Pichugina et al. (2019), Collins et al. (2024), Turner et al. (2022), Ghate et al. (2022). Given the scarcity of hub-height observations (see Introduction), HRRR provides flexible samples at random locations to emulate in-situ format for demonstrating our workflow and evaluating `WindSR`.

because WTK and HRRR have slightly different spatial extents and resolutions, we first crop WTK to HRRR's coverage, then regrid HRRR to the WTK grid (2km grid spacing) to align datasets for training and DA. Due to the large data volume (>5 million grid cells), the entire domain is partitioned into $128 \times 128$-cell patches. All experiments are conducted at the patch level.

**Critical Data Needs and Limitation** Ideally, hub-heights wind speed and direction would be measured across the U.S., as near-surface ASOS data are, but only 25 sites currently provide 13-100m winds to the public. Using these sparse, nonuniform heights in ML models is difficult without site-specific, multi-height training. Moreover, our dynamic-impact-radius assimilation scheme requires nearby observations, which are typically unavailable at hub-height. HRRR, continually improved and validated through their collaboration with DOE Wind Energy Technology office (e.g., Ghate et al. (2022)), performs well over land and offshore in both calm and extreme conditions, enabling us to demonstrate our framework and support user applications.

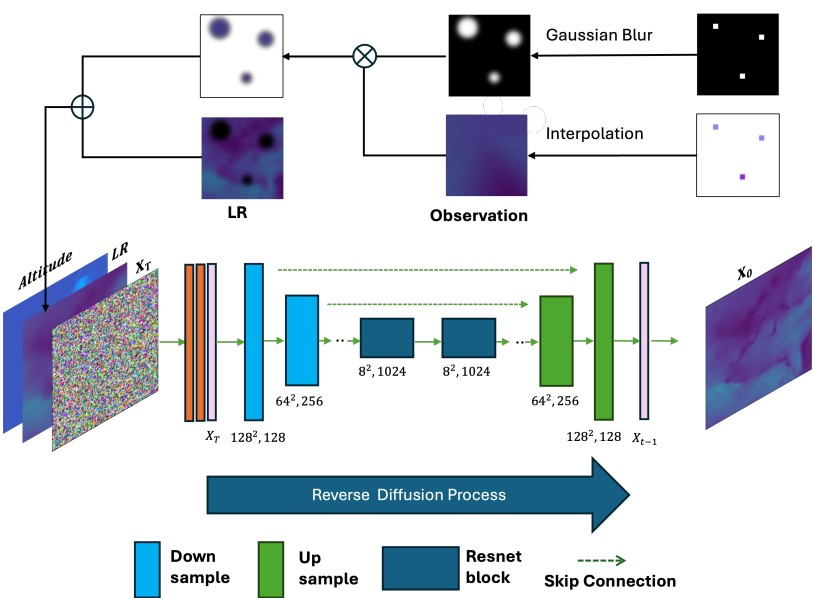

Figure 3: Image-generation workflow combining DA and SR to downscale windspeed. Sparse observations are interpolated to the inference grid and softly blended via a mask, then inpainted into the simulation filed to form a composite that conditions the diffusion model during generation; terrain information is included as an additional conditioning input during the reverse process.

## 4 METHOD

### 4.1 OVERVIEW

This work integrates DA into the diffusion-based downscaling workflow by inpainting sparse observations into low-resolution images that then serve as conditioning inputs for the diffusion model to denoise high-resolution outputs. We employ SR3 as the primary SR tool. During training, terrain information is included as an additional conditioning variable alongside the initial condition $x_t$ and the low-resolution image. During inference, to ensure sparse observations effectively influences the result, we dynamically adjust each observation's impact radius (via a Gaussian kernel) in the denoising process. This design accounts for terrain and local wind-speed variability around observation points, optimizing the use of observations.

### 4.2 CONDITION IN TRAINING

Over complex terrain, winds vary rapidly and remain challenging to simulate in numerical models even at very high spatial resolutions, with substantial computational cost Ghate et al. (2022); Dettling et al. (2025). This motivates using terrain as conditioning information in the diffusion model.

Our backbone for super-resolution is SR3 with a U-Net architecture as shown in Figure 3. SR3 starts from Gaussian noise $x_t$ and conditions on a low-resolution image. We enhance this by adding terrain as an extra channel (128x128) to the conditioning tensor, concatenated with the interpolated low-resolution image. The U-Net then denoises $x_t$ given this conditioning. Throughout this paper, this enhanced SR3 serves as our pretrained downscaling model.

### 4.3 DATA ASSIMILATION IN INFERENCE

Following similar inpainting techniques in Lugmayr et al. (2022); Huang et al. (2024); Song et al. (2020), we insert observations into the conditioning image prior to inference. To emulate sparse observations, HRRR data are sampled at different weather stations location on the map; within each 128x128 patch (2km WTK resolution makes 256 km x 256 km areas), we use one or multiple points. These points represent real-world station observations. We interpolate these sparse observation points to the simulation grid, then generate a soft mask that defines a support area (details are showed in Figure 3).

Adapting the soft-mask method of Huang et al. (2024) Huang et al. (2024), we blend observations into the WTK field using a Gaussian kernel soft-bleed function. A key innovation is the dynamic impact radius $d$, determined by terrain variance and surrounding WTK wind speed variance: over complex terrain, where winds fluctuate rapidly, $d$ is limited; over flat terrain, $d$ can expand. Previous work Huang et al. (2024) treated each observation point as having equal influence on the simulation. Operationally, we increase $d$ by 1 pixel at each step, computing terrain and wind-speed variance within the covered area; once a threshold in variance is exceeded, expansion stops and the final $d$ (bounded 1-6 pixels) is assigned. Algorithm 1 outlines this dynamic adjustment procedure.

$$x = (m_{weight} \odot x_{\text{HRRR}}) + ((1 - m_{weight}) \odot x_{\text{WTK}}) \tag{4}$$

Once $d$ is determined, the kernel masks portions of the interpolated observation image, which are then inpainted into the simulation image for downscaling. This composite image is defined in Equation 4, where $m_s$ represents the weight of the observation values through the soft-blend function, $x_{HRRR}$ is the observation image with sparse point, and $x_{WTK}$ is the simulation image. The resulting composite image, $x$, is then passed to the pretrained `WindSR` downscaler, which conditions on terrain and initiates from pure noise. The output is a high-resolution image with assimilated observations, enhancing detail and accuracy relative to the input low-resolution simulation data.

### 4.4 LIMITATION AND FUTURE EXTENSION

Winds exhibit strong spatial and temporal variability. This study focused on spatial SR, However, temporal conditioning, such as wind speed data from adjacent time steps, could be incorporated in future work to generate high-resolution winds in both space and time, which is important for accurately quantifying wind gusts. In addition to terrain information, land use/land cover as well as air temperature can also be incorporated, subject to data availability to users.

---

**Algorithm 1** Dynamic Impact Radius

---

**Input:** Coordinate $p$, terrain deviation threshold $T_1$, wind speed deviation threshold $T_2$

1: **procedure** DIR$(p, T_1, T_2)$
2:     Define $min\_radius \leftarrow 1, max\_radius \leftarrow 6$
3:     Initialize the impact radius $r \leftarrow min\_radius$
4:     **while** $r < max\_radius$ **do**
5:         $a = \{(x, y) \in \mathbb{R}^2 \mid (x - p_x)^2 + (y - p_y)^2 \leq r^2\}$
6:         $\sigma_h = \sqrt{\frac{1}{N} \sum_{\hat{p} \in a} (h_{\hat{p}} - \bar{h})^2}$ , $h_{\hat{p}}$ : terrain height at point $\hat{p}$
7:         $\sigma_s = \sqrt{\frac{1}{N} \sum_{\hat{p} \in a} (s_{\hat{p}} - \bar{s})^2}$ , $s_{\hat{p}}$ : wind speed at point $\hat{p}$
8:         **if** $\sigma_h < T_1$ **and** $\sigma_s < T_2$ **then**
9:             $r \leftarrow r + 1$
10:        **else**
11:           *break*
12:        **end if**
13:     **end while**
14:     **return** the impact radius $r$
15: **end procedure**

---

## 5   WindSR Experiments and Evaluations

We conduct experiments to assess the effectiveness of our diffusion-based SR and DA-enabled processes, comparing them with established SR models. The configuration of WindSR is detailed in subsection 5.1; comparisons and evaluations appear in subsection 5.2.

### 5.1   Implementation

Our WindSR model is trained on 10,000 images of size 128x128 at 2 km resolution, randomly sampled from 2018 WTK winds across entire Contiguous United States. For training, images are downsampled (or coarsened) to 16x16 (16 km) to form paired low-/high-resolution inputs. Training is conducted on a single node with four A100 GPUs for 300,000 iterations around 140 hours, following the SR3 default settings. Inferences uses a single A100 GPU, with around 25 seconds per image.

To evaluate WindSR against prior models, we also train SRCNN and ESGAN on identical 2 km targets and upsampled 16 km inputs from 2018 WTK. SRCNN is a lightweight CNN with shallow depth; ESRGAN employs an adversarial framework to produce sharper details. When training ESRGAN, we encountered convergence issues due to a high content-loss weight (tuned for images with sharp contours). This approach does not necessarily work for wind fields, so we reduce the content-loss weight and increase pixel-wise loss, enabling ESRGAN to converge and capture fine-grained structure. We also evaluate the impact of DA within WindSR. Because both SRCNN and ESRGAN aim to minimize pixel-wise errors on wind values (typically MSE or MAE), they can yield overly smooth wind fields. In contrast, diffusion-based WindSR aims to minimize a noise-prediction loss conditioned on low-resolution input and terrain, helping recover fine-scale texture — an advantage over purely CNN-based SR models. In terms of evaluations, because our training involves gridded WTK data and sparse (emulated) observations from HRRR, we evaluate against HRRR rather than WTK to reflect observations-anchored performance.

### 5.2   Comparison of Super Resolution Models

We compare SR performance using Table 2, which reports mean Structural Similarity Index (SSIM) and Peak Signal-to-Noise Ratio (PSNR) over 200 images (referenced to 2 km WTK for this diagnostic). Larger SSIM and PSNR indicate better agreement with the reference. The terrain-aware WindSR attains the highest PSNR and SSIM. In contrast, SRCNN achieves the lowest PSNR and SSIM

|  | PSNR ($\uparrow$) | SSIM ($\uparrow$) |
|---|---|---|
| SRCNN Dong et al. (2015) | 27.34 | 0.7005 |
| ESRGAN Wang et al. (2018) | 28.16 | 0.7021 |
| WindSR w/o Terrain | 31.69 | 0.8105 |
| **WindSR w/ Terrain** | **32.83** | **0.8207** |

Table 2: The average SSIM and PSNR across 200 random sampled images for various models.

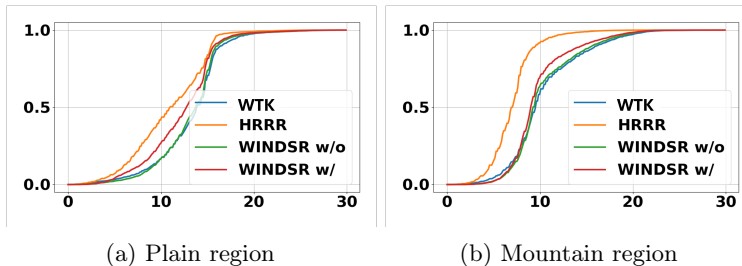

(a) Plain region        (b) Mountain region

Figure 4: Cumulative distribution functions of wind speeds across different terrain conditions of the US, comparing values recorded by HRRR, WTK, and our customized SR with and without DA. In both flat and mountainous regions, DA brings the distribution of wind speeds closer to HRRR data which is used to emulate observations.

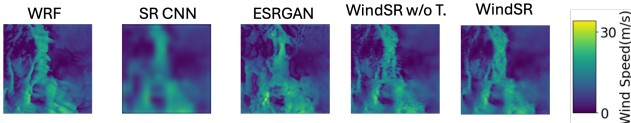

Figure 5: Visual comparison of different SR models

because when they try to achieve small model errors, they tend to produce smoother fields and miss fine-scale features, evident in the WTK original image (see Figure 5). The WindSR model without terrain still outperforms SRCNN and ESRGAN, suggesting that diffusion-based SR (which targets noise-space learning) better preserves detail than models minimizing only pixel-space loss. For visual comparison, Figure 5 shows a representative 128x128 tile (256 km x 256 km): SRCNN under-resolves small-scale structure despite its relatively acceptable statistics; ESRGAN is sharper with similar statistics to SRCNN; terrain-conditioned WindSR excels in both statistics and visual fidelity.

### 5.3 Evaluation of Data Assimilation

We examine pixel-level differences between HRRR (observations) and outputs from our diffusion-based SR model with and without DA, and we test the proposed dynamic-radius design against fixed-radius settings. Table 3 summarizes Mean Absolute Error (MAE) and Root Mean Square Error (RMSE) for SRCNN, ESRGAN, WindSR (with and without terrain) across multiple fixed radii and the dynamic-radius case. WindSR with DA conditioning performs the best; with a dynamic impact radius it achieves the lowest MAE (1.64) and RMSE (2.01), indicating that dynamically adapting the influence radius improves accuracy. We further illustrate DA effects using cumulative distribution functions of wind speed across U.S. terrain classes Figure 4. In both terrain types, all models tend to underestimate portions of the distribution – especially in mountainous regions, where SR distributions are generally lower than HRRR across most quantiles. DA shifts the `WindSR` distributions closer to HRRR (serves as observational reference), particularly at higher wind speeds (greater than 10 m/s), where density is improved. This is encouraging, as SR models commonly struggle in complex terrain and at large wind speeds. Spatial error patterns in Figure 6 show that `WindSR` with DA shows lower RMSEs —3.173, 5.176 m/s, and 3.757 m/s —than the no-DA runs (4.025 m/s, 6.425 m/s, and 4.834 m/s, corresponding to bias reduction of 21.17%, 19.44%, and 22.28% over three representative tiles, highlighting the effectiveness of DA in reducing simulation errors.

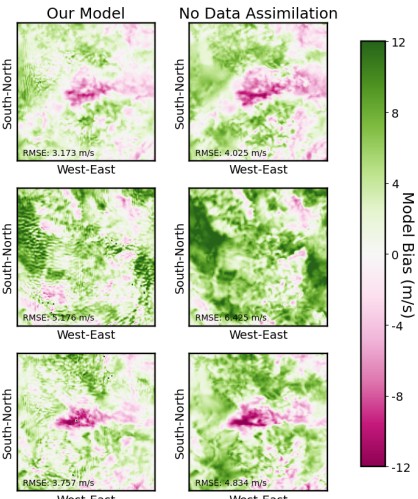

Figure 6: Comparison between our model with data assimilation vs. no data assimilation in the same location. The images exhibit minimal model bias, indicated by white or near-white colors, the lighter the color, the less bias.

We further assess the effectiveness of DA using real-world weather station observations. The selected area is located in Illinois, U.S., with a latitude span from 38.671° to 41.275° and a longitude span

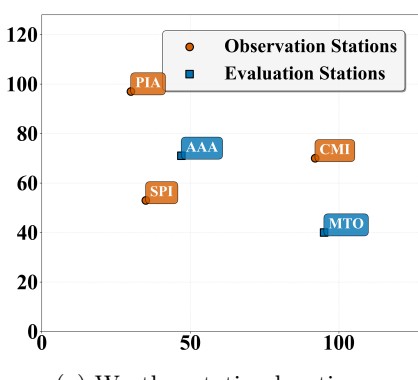

(a) Weather station locations

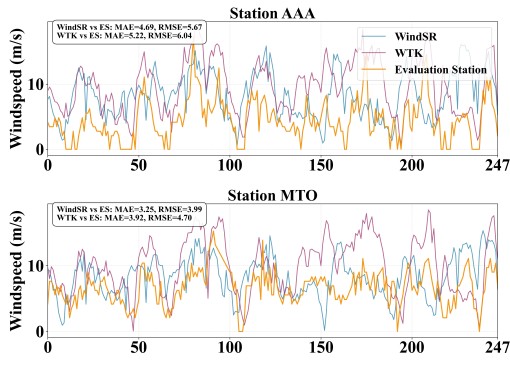

(b) Station AAA and MTO windspeed change

Figure 7: (a) Displays the locations of the weather stations within the selected region and their respective usage. (b) Shows the wind speed changes over the one-month period, along with the corresponding MAE and RMSE values.

| | Radius 2 | | Radius 4 | | Radius 6 | | Dynamic | |
|---|---|---|---|---|---|---|---|---|
| | MAE (↓) | RMSE (↓) | MAE (↓) | RMSE (↓) | MAE (↓) | RMSE (↓) | MAE (↓) | RMSE (↓) |
| SRCNN Dong et al. (2015) | 1.78 | 2.17 | 1.70 | 2.11 | 1.69 | 2.10 | 1.69 | 2.09 |
| ESRGAN Wang et al. (2018) | 1.92 | 2.34 | 1.82 | 2.25 | 1.80 | 2.23 | 1.81 | 2.24 |
| WindSR w/o Ter. | 1.81 | 2.20 | 1.74 | 2.15 | 1.72 | 2.13 | 1.71 | 2.12 |
| **WindSR** | **1.76** | **2.15** | **1.70** | **2.08** | **1.68** | **2.12** | **1.64** | **2.01** |

Table 3: Different models with different radius comparison.

from -97.240° to -90.639°, covering a $256 \times 256$km region, which corresponds to a $128 \times 128$ size patch at 2km resolution. We identified five weather stations in this area, three of which serve as observation stations (PIA, SPI, CMI) and two as evaluation stations (AAA, MTO) as shown in Figure 7a. Our goal is to quantify the error correction at the evaluation stations after data assimilation with the observation station data, so that to verify the effectiveness of the WindSR method.

Since these stations only provide 10m wind measurements, we use a power-law approach Jung & Schindler (2021) to infer 80m wind speeds. The weather station data and WTK data span one month, from January 1 to February 1, 2018, with 3-hourly intervals, totaling 248 time steps. By using the WindSR approach and inpainting the data from the three observation stations into the WTK dataset to create the initial conditions for the SR model, we generate new high-resolution data. We compare these results with the WTK simulation data and the real observations at the two evaluation stations. As shown in Figure 7b, WindSR effectively reduces both the MAE and RMSE relative to the real observations at both stations. At station AAA, the MAE decreases from 5.22 to 4.69, and the RMSE decreases from 6.04 to 5.67. At station MTO, the MAE decreases from 3.92 to 3.25, and the RMSE decreases from 4.7 to 3.99. Overall, the bias is reduced by approximately 10–20%.

## 6 CONCLUSION

This paper presents WindSR, a diffusion-based wind SR framework that integrates sparse observational data with rich gridded simulations. We introduce a dynamic impact-radius scheme that adaptively blends observations with simulations, providing spatially adaptive conditioning for the diffusion process. Terrain information is incorporated during both training and inference to enhance physical realism. We evaluate WindSR against CNN- and GAN-based baselines in terms of downscaling performance and data assimilation effectiveness on both simulation data and weather station observations. WindSR outperforms these methods, achieving higher accuracy and more effectively integrating observational data with simulation data, thereby efficiently correcting simulation biases.

## 7 LLM USAGE

We used a large language model (LLM) solely to improve the writing, including grammar, clarity, and readability. The LLM did not contribute to research ideation, methodology, analysis, or results. All scientific content and conclusions remain the sole responsibility of the authors.

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
