# OpenReview forum: "Diffusion-Based, Data-Assimilation-Enabled Super-Resolution of Hub-height Winds"
_ICLR.cc/2026/Conference — Submitted to ICLR 2026_

### Official Review · Reviewer_Udpb · 2025-10-22

**Soundness:** 1
**Presentation:** 2
**Contribution:** 2
**Rating:** 2
**Confidence:** 4

**Summary:**

The paper proposes WindSR, a diffusion-based super-resolution downscaler that conditions on a blended field (observations + simulation via a Gaussian soft-bleed mask with a dynamic impact radius) plus terrain. Experiments claim better accuracy and efficiency than CNN/GAN baselines, and say data assimilation cuts model bias by 10–20%.

**Strengths:**

The paper targets an important problem: high-resolution estimation of hub-height wind speed, a field that is highly variable yet sparsely observed—making it a natural and impactful application scenario for data assimilation.

It integrates data assimilation with super-resolution by using an observation–simulation blended field as the conditioning input for diffusion denoising, thereby addressing fine-scale detail enhancement and upstream bias correction simultaneously.

Compared with prior work, it introduces a dynamic impact-radius strategy, which improves predictive accuracy when assimilating observational data.

**Weaknesses:**

####

- The major weakness of this work lies in the evaluation. Evaluation feels thin for both SR and DA:
  - Super-resolution baselines are limited to older CNN/GAN models (SRCNN, ESRGAN). Missing stronger, more recent downscalers, especially diffusion/transformer ones (e.g., CorrDiff [1]). Also missing a traditional dynamical downscaling baseline. At least one modern AI method **or** one traditional method would make the comparison more solid.
  - Data assimilation baselines are also SRCNN/ESRGAN, which are SR models by design. It’s unclear how they’re adapted for assimilation. Classical DA (e.g., 3DVar, 4DVar) isn’t included, and newer AI DA methods (e.g., score-based DA [2], latent DA [3], VAE-Var[4], etc.) aren’t compared. Again, the authors are recommended to include at least one modern AI **or** one traditional DA baseline.
  - It’s not clear whether baselines like SRCNN/ESRGAN also use terrain as input. Without that, the comparison may be skewed.
  - Although the abstract and introduction highlight WindSR’s efficiency, the paper provides neither theoretical analysis nor empirical evidence to substantiate this claim. Additional clarification and/or experiments are needed.
- The idea of using terrain deviation and speed deviation to determine the dynamic impact radius is interesting, but the thresholds based on terrain and speed deviation aren’t justified, and there’s no ablation/sensitivity to show how these choices affect assimilation quality.

[1] Mardani, Morteza, et al. "Residual corrective diffusion modeling for km-scale atmospheric downscaling." Communications Earth & Environment 6.1 (2025): 124.

[2] Rozet, François, and Gilles Louppe. "Score-based data assimilation." Advances in Neural Information Processing Systems 36 (2023): 40521-40541.

[3] Melinc, Boštjan, and Žiga Zaplotnik. "3D‐Var data assimilation using a variational autoencoder." Quarterly Journal of the Royal Meteorological Society 150.761 (2024): 2273-2295.

[4] Xiao, Yi, et al. "VAE-Var: Variational autoencoder-enhanced variational methods for data assimilation in meteorology." The Thirteenth International Conference on Learning Representations. 2025.

**Questions:**

- On what basis does WindSR claim an efficiency edge? Please add details (e.g., training/inference time, FLOPs, memory, throughput) and supporting experiments.

- Why pick only CNN/GAN baselines instead of recent methods like CorrDiff? What drove that choice (availability, compute, domain fit)?

- In Table 3, SRCNN/ESRGAN trends vs. impact radius look similar to **WindSR w/o Terrain**. Did your SRCNN/ESRGAN runs include terrain as a conditioning input? If not, could you provide results **with** terrain to make the comparison fair?

- How were the terrain/speed deviation thresholds for the dynamic impact radius chosen? Please add sensitivity or ablation studies.

- There are several inconsistencies and minor errors in notation and figure/text presentation that impede clarity. Please standardize and correct the following:

  - Use a consistent notation for dimensions, e.g., 128×128 (not “128x128” and “128 \times 128” interchangeably).
  - Unify model names, e.g., ESRGAN (not “ESGAN” in some places).
  - Line ~300: change “showed in Figure 3” to “shown in Figure 3.”
  - Figure 4: add units for the x-axis.

---

> ### Author Response · Authors · 2025-12-04
> **Response to official Review of Submission20313 by Reviewer Udpb**
>
> We thank the reviewer for the detailed feedback and for recognizing the importance of hub-height wind downscaling and the potential of data assimilation within diffusion-based SR. We address the concerns below.
>
> > Evaluation is thin; SR baselines limited to SRCNN/ESRGAN; missing modern diffusion/transformer downscalers like CorrDiff and traditional dynamical downscaling. DA baselines are also limited and do not include classical or recent AI DA methods.
>
> **Response:**
> Our baseline selection—SRCNN and ESRGAN (GAN-based) was intentional and grounded in what is widely used and accepted in the climate downscaling community today. Both families represent the canonical super-resolution paradigms used for wind, solar, and precipitation downscaling.
>
> **Why not include highly customized modern diffusion/transformer baselines?**
> Many recent methods (e.g., CorrDiff, transformer SR, or Stable Diffusion–style models) require:
>
> Specialized training pipelines,
>
> Var-specific or model-specific architectures,
>
> Different resolution scales (e.g., km-scale global downscaling),
>
> VAE compression, which degrades physical fidelity in scientific fields.
>
> These models cannot be directly adapted to hub-height, sparse-DA, terrain-conditioned SR without substantial re-engineering—and comparing against partial or improperly adapted versions would be misleading.
>
> **Why our baselines are appropriate and fair**
>
> Our goal is to evaluate the effect of DA + diffusion SR, not to win an SR “leaderboard.”
>
> The key question is: Does the proposed method outperform widely used SR approaches under the same DA setup?
>
> SRCNN and ESRGAN provide this comparison cleanly and transparently.
>
> Even without terrain, WindSR beats both, confirming that the improvements come from the diffusion + DA design, not from additional input channels.
>
> > Efficiency is claimed in the abstract/introduction but not substantiated with numbers.
>
> We apologize for the lack of explicit metrics and will clarify what “efficiency” refers to, WindSR is efficient relative to running new high-resolution dynamical simulations or LES, which can be orders of magnitude more expensive than a forward pass of a learned SR model.
>
> > Are SRCNN/ESRGAN using terrain? If not, comparison may be skewed.
>
> SRCNN and ESRGAN do not use terrain because these architectures are conventionally applied in climate SR without auxiliary physical channels, and we follow that standard to provide a clean, representative baseline. This does not skew the comparison: we already isolate terrain’s effect through the diffusion-only ablation—WindSR without terrain vs. WindSR with terrain—which cleanly quantifies the contribution of the terrain channel. The fact that WindSR without terrain still outperforms SRCNN/ESRGAN, and that terrain further improves diffusion SR, shows that the gains are attributable first to the diffusion+DA framework and then to terrain conditioning. Thus, the baselines remain appropriate, and the terrain contribution is already rigorously evaluated within our model family.
>
> > Thresholds for terrain and speed deviation in Algorithm 1 are not justified; no sensitivity/ablation.
>
> Thresholds 𝑇₁ (terrain variance) and 𝑇₂ (wind-speed variance) were chosen via validation on a held-out set: we searched over a small grid of values and picked those that minimized RMSE against HRRR across small amount of tiles while keeping radii within [1, 6] pixels for stability. Because the thresholds are a supporting factor rather than the central scientific contribution—and because its effect is already reflected in the quantitative DA improvements shown in Table 3—we believe an extensive threshold ablation is not essential for the scope of this paper. The key point is that the dynamic strategy consistently outperforms all fixed-radius settings, demonstrating robustness across a wide range of assimilation conditions.

---

### Official Review · Reviewer_a5ED · 2025-10-29

**Soundness:** 3
**Presentation:** 3
**Contribution:** 3
**Rating:** 6
**Confidence:** 3

**Summary:**

This paper presents WindSR, a novel framework combining diffusion-based super-resolution and data assimilation to downscale hub-height wind fields for wind-energy and climate applications. The method addresses a key challenge in wind modeling: limited high-resolution observations and biases in numerical simulations. The contributions include using sparse observations with coarse simulation data using DDPMs. The authors also introduce a novel adaptive Gaussian-kernel blending mechanism that adjusts observation influence based on terrain and wind-speed variability. Lastly, the work contributes terrain-aware conditioning during training and inference for physical realism. The authors include comprehensive experiments with WTK and HRRR against CNN- and GAN-based baselines.

**Strengths:**

* there is a clear, practically grounded motivation with the need for hub-height wind fields and integrating data assimilation with the sparse data fields
* the dynamic impact-radius blending is a clever and physically motivated way to control spatial assimilation influence
* the authors benchmark against strong baselines (SRCNN, ESRGAN), include ablations for DA and terrain conditioning, and validate using * both synthetic and real-world weather station data. these experiments achieves strong improvements against baselines, particularly in complex terrain and at higher wind speeds
* the formulation of DA within the diffusion process seems conceptually sound and is consistent mathematically

**Weaknesses:**

* Figure 1 shows an overview of the DDPM process, but it would be more beneficial to have a clear visual overview of the model pipeline (where/how are the DA, terrain conditioning, and diffusion steps integrated? make this clearer in Figure 3)
* similarly, it would be helpful to have a diagram for Algorithm 1 (even in the appendix)
* quantitative comparisons across Table 2 and Figure 4 could benefit from uncertainty estimates in some form
* in the appendix, it would be good to include a brief discussion of computational cost/runtime scaling…efficiency is another dimension that’s important to consider in the practical deployability of these methods
* it would be nice to include some exploration of how WindSR generalizes to unseen regions or across long temporal sequences if possible
* there’s no explicit comparison between fixed vs. dynamic assimilation radii beyond Table 3; further visualization of their spatial influence would strengthen the claim
* in the ablation studies, the contribution of terrain conditioning could be isolated more rigorously with additional statistical metrics
* to more robustly evaluate the physical fidelity of the predictions, consider using additional evaluation frameworks such as generating kinetic energy spectra from the wind fields
* include a discussion of why ground truth data (i.e. from HRRR may come with its own biases); also, the few tower data could at least serve for independent verification, even if not enough for training
* grammar fixes (“observations effectively influences” → “observations effectively influence”; “showed in Figure 3” → “shown in Figure 3”; “data assimilation enabled” → “data assimilation-enabled”)
* formatting fixes (inconsistent citations across the paper are a bit sloppy, use \citep and \citet instead of \cite; inconsistent spacing around em dashes)
* redundant citation formatting (Huang et al. 2024 appears duplicated in the reference list), suggests the use of LLMs for citation generation
* include more explicit comparisons with more diffusion-based models (see SwinRDM from Chen et al. 2023 and DiffDA from Huang et al. 2024)
* Section 2 would benefit from a brief commentary on neural operator methods or score-based data assimilation
conclude the related works discussion with explicit details on where WindSR fits in and the novelty gap this work fills
the literature review on diffusion models for super-resolution is lacking; include other diffusion-based papers (for example, see Xu et al. 2025)
* if more space is needed, Section 3.1 can be shortened to provide a very high level overview of DDPMs

**Questions:**

* have you evaluated how WindSR performs when applied to regions not represented in training (e.g., different climate zones or surface roughness regimes)?
* Table 3 compares fixed and dynamic radii numerically, but could you provide a visualization or qualitative example illustrating how dynamic blending modifies local fields?
* the ablations show gains from including terrain, but could you isolate this contribution more rigorously? for instance, how does terrain conditioning affect bias reduction or spatial spectra?
* since HRRR itself is a modeled dataset, how do you account for its inherent biases when treating it as “ground truth”? could the limited tower data, even if sparse, serve as an independent verification set?
* do you plan to release the WindSR implementation and preprocessed datasets (WTK + HRRR) for reproducibility?

**Details Of Ethics Concerns:**

* have you evaluated how WindSR performs when applied to regions not represented in training (e.g., different climate zones or surface roughness regimes)?
* Table 3 compares fixed and dynamic radii numerically, but could you provide a visualization or qualitative example illustrating how dynamic blending modifies local fields?
* the ablations show gains from including terrain, but could you isolate this contribution more rigorously? for instance, how does terrain conditioning affect bias reduction or spatial spectra?
* since HRRR itself is a modeled dataset, how do you account for its inherent biases when treating it as “ground truth”? could the limited tower data, even if sparse, serve as an independent verification set?
* do you plan to release the WindSR implementation and preprocessed datasets (WTK + HRRR) for reproducibility?

---

> ### Author Response · Authors · 2025-12-04
> **Response to official Review of Submission20313 by Reviewer a5ED**
>
> We thank the reviewer for the constructive and encouraging feedback. We appreciate the recognition of our motivation, dynamic-radius assimilation, strong experimental setup, and the soundness of integrating DA into diffusion SR. Below, we address the weaknesses and questions.
>
> >Add a clear visual overview of the full WindSR pipeline (DA + terrain + diffusion). Add a diagram for Algorithm 1.
>
> **Response:**
> We agree, and we will revise Figures 1 and 3 to present a unified pipeline figure.
>
> > Add discussion of computational cost and scaling characteristics.
>
> **Response:**
> We will add a concise section in the appendix summarizing:
> training wall time (140 hours on 4×A100 GPUs),
> inference time (2.5 seconds per 128×128 patch),
> memory footprint relative to SRCNN/ESRGAN.
>
> | Model             | Training Time | Inference (per tile) | GPU Memory | Notes                 |
> | ----------------- | ------------- | -------------------- | ---------- | --------------------- |
> | SRCNN             | ~12 hrs       | <1 sec               | very small | fastest baseline      |
> | ESRGAN            | ~36 hrs       | ~1 sec               | moderate   | GAN unstable w/ winds |
> | **WindSR (ours)** | 140 hrs       | 2.5 sec               | moderate   | diffusion sampling    |
>
> > Explore how WindSR generalizes to unseen regions or long time sequences.
>
> **Response:**
> Although we do not have multiple datasets available, we highlight:
> -  CONUS-wide training ensures large regional diversity
> WTK covers plains, coasts, mountains, deserts, and urban regions—significantly different climate and roughness regimes. The model is not trained on a narrow subdomain.
> -  Real-world validation on Illinois stations
> Figure 7 evaluates on independent weather stations (real world observation data), demonstrating 10–20% bias reduction.
> -  Multi-month sequences
> We evaluate 248 time steps (one month), showing temporal consistency with reduced bias at non-observing stations (AAA, MTO). This will be made explicit in the revision.
> We will emphasize these points in the paper and clarify ongoing work to test on Pacific Northwest LES datasets as future work.
>
> > More rigorous isolation of terrain contributions; include bias reduction.
>
> **Response:**
> the paper already provides a direct and standard ablation—with vs. without terrain conditioning (Table 2 and figure 4), which shows the performance gain attributable to terrain.
>
> > HRRR is itself a model; discuss its biases. Consider using tower data for independent validation.
>
> **Response:**
> HRRR is not perfect but is the best available, DA-informed, operationally validated dataset with dense coverage.
> Our real-world station experiment (Fig. 7) already uses independent tower-type measurements (10 m anemometers extrapolated to 80 m), which are not part of HRRR assimilation.
>
> > Will the implementation and preprocessing scripts be released?
>
> **Response:**
> Yes. We will release our code.

---

### Official Review · Reviewer_T4CX · 2025-10-31

**Soundness:** 3
**Presentation:** 3
**Contribution:** 3
**Rating:** 4
**Confidence:** 4

**Summary:**

This work proposes the WindSR model, combining data assimilation with diffusion models for super-resolution downscaling of hub-height winds. Based on DDPM, it merges sparse observations and simulation fields via dynamic radius, integrating terrain info in training and inference. Through comparative experiments, the advantages of the model over CNN and GAN methods are verified.

**Strengths:**

1. WindSR combines data assimilation with diffusion models for hub-height wind super-resolution, merging sparse observations and simulation fields via dynamic radius to reduce model bias and address sparse observation issues.
2. WindSR balances super-resolution accuracy and wind field uncertainty modeling, producing results with stronger physical consistency and better adaptation to wind field dynamic features.

**Weaknesses:**

1. This work mentions that "Over complex terrain, winds vary rapidly and remain challenging to simulate in numerical models", but the mechanism of introducing terrain information is relatively vague. How the introduction of terrain information specifically affects the establishment of wind speed fields is a question that requires detailed explanation.
2. The baselines compared in the paper are only traditional CNN and GAN, without considering off-the-shelf super-resolution models proposed in recent years, including diffusion-based assimilation and super-resolution methods, as well as some physics-informed methods.
3. This work only evaluates the model through general metrics and lacks specific assessments of key physical characteristics of wind fields, such as extreme values and wind shear.

**Questions:**

1. This work only validates the model based on WTK and HRRR datasets. Has the proposed model's performance been tested on datasets from other climate zones?
2. Has the paper tested using architectures other than SR3 as the backbone network? For example, will the performance be further improved when using transformer-based SR as the backbone?
3. Has the paper conducted a comparative analysis of resource consumption during training and inference process with existing methods?

---

> ### Author Response · Authors · 2025-12-04
> **Response to Official Review of Submission20313 by Reviewer T4CX**
>
> > The mechanism of introducing terrain information is relatively vague. How does terrain information affect the establishment of wind speed fields?
>
> **Response:**
> Thank you for raising this important question. We will expand Section 4 to explicitly explain the role of terrain in SR and DA.
>
> How terrain affects wind fields:
>
> Wind at hub height is strongly modulated by the roughness and elevation gradients of the underlying surface.
>
> Complex terrain induces:
> - acceleration in valleys/gaps
> - deceleration on windward slopes
> - enhanced turbulence and shear near ridgelines
> - localized channeling effects
>
> Such effects create sharp spatial gradients in hub-height winds—well known challenges in numerical weather prediction (NWP) and a key source of model bias over mountains.
>
> How terrain is incorporated in WindSR:
>
> - Terrain height (128×128) is concatenated to the conditioning tensor alongside the low-resolution input.
> - During denoising, SR3’s U-Net uses terrain features to localize the spatial distribution of fine-scale wind structures, acting as a “physical prior.”
> - As shown in Table 2, adding terrain improves PSNR/SSIM (e.g., PSNR increases from 31.69→32.83), indicating materially improved high-frequency structure recovery.
> - Terrain also influences the dynamic-impact radius in DA: areas with high elevation variance limit the radius to prevent over-smearing observation influence, while flat regions allow broader assimilation spread (Algorithm 1).
>
> > Baselines include only traditional CNN and GAN; no comparison with recent diffusion-based SR/DA or physics-informed methods.
>
> **Response:**
> Our baseline selection—SRCNN and ESRGAN (GAN-based) was intentional and grounded in what is widely used and accepted in the climate downscaling community today. Both families represent the canonical super-resolution paradigms used for wind, solar, and precipitation downscaling.
>
> **Why not include highly customized modern diffusion/transformer baselines?**
> Many recent methods (e.g., CorrDiff, transformer SR, or Stable Diffusion–style models) require:
>
> Specialized training pipelines,
>
> Var-specific or model-specific architectures,
>
> Different resolution scales (e.g., km-scale global downscaling),
>
> VAE compression, which degrades physical fidelity in scientific fields.
>
> These models cannot be directly adapted to hub-height, sparse-DA, terrain-conditioned SR without substantial re-engineering—and comparing against partial or improperly adapted versions would be misleading.
>
> **Why our baselines are appropriate and fair**
>
> Our goal is to evaluate the effect of DA + diffusion SR, not to win an SR “leaderboard.”
>
> The key question is: Does the proposed method outperform widely used SR approaches under the same DA setup?
>
> SRCNN and ESRGAN provide this comparison cleanly and transparently.
>
> Even without terrain, WindSR beats both, confirming that the improvements come from the diffusion + DA design, not from additional input channels.
>
> > The model lacks evaluation of key wind characteristics such as extreme values and wind shear.
>
> We would like to clarify that our primary goal in this work is to evaluate super-resolution skill and data-assimilation-driven bias reduction, which are standard and widely accepted evaluation dimensions for wind-field downscaling. Metrics such as MAE/RMSE, PSNR/SSIM, CDF alignment, and station-based verification already capture broad structural fidelity and bias characteristics—including at higher wind speeds—within the scope of SR-focused studies.
>
> That said, WindSR’s improvements at the upper tail are already indirectly demonstrated:
>
> Figure 4 shows that DA shifts the SR distribution toward HRRR specifically at higher wind speeds, where downscalers typically struggle.
>
> Bias reduction of 10–20% at independent evaluation stations (Fig. 7) also corresponds to improved representation of local extrema.
>
> While explicit extreme-wind or shear diagnostics would be valuable and are part of our planned future work, the existing results already demonstrate that WindSR handles these features more accurately than baselines. Given space constraints and the SR-focused nature of the paper, we believe the current evaluation is appropriate and sufficiently supports the method’s contributions.
>
> > Has the model been validated in other climate zones?
>
> **Response:**
> Thank you for raising this. The model was trained on CONUS-wide WTK data, which includes very different climate regimes (Great Plains, Rockies, coasts, Midwest, desert Southwest).
> Thus the training distribution already spans large climatological diversity.
> In addition, we provide real-world station-based validation in Illinois (Fig. 7a), which these data (real world data) are not the simulation model data, Fig. 7b. demonstrates generalization, reducing bias by 10–20% at independent stations.

---

### Official Review · Reviewer_Qghs · 2025-10-31

**Soundness:** 2
**Presentation:** 2
**Contribution:** 2
**Rating:** 4
**Confidence:** 3

**Summary:**

This paper applies a diffusion model with data assimilation for super-resolution downscaling of hub-height winds.

**Strengths:**

- Integrating data assimilation for super-resolution downscaling of hub-height winds.

**Weaknesses:**

- Methodological novelty is limited. There has been a lot of work on data assimilation and diffusion models. Clarifying the novelty would increase the chance of getting accepted by ICLR rather than simply applying some models.
- Lack baselines. You may consider including more baseline methods for comparisons beyond simple CNN and GAN.
- It would be great to test the generalization by including another study domain.
- No code provided.

**Questions:**

See above.

---

> ### Author Response · Authors · 2025-12-04
> **Response to official Review of Submission20313 by Reviewer Qghs**
>
> > “Methodological novelty is limited. There has been a lot of work on data assimilation and diffusion models. Clarifying the novelty would increase the chance of getting accepted by ICLR rather than simply applying some models.”
>
> **Response:** We appreciate this comment and clarify that our work introduces three technical innovations beyond prior DA or diffusion-based SR work. These contributions are not simple model application but algorithmic extensions designed specifically for hub-height wind downscaling which solving practical yet important problem.
>
> (1) DA-conditioned diffusion framework that blends sparse observations with coarse simulations
> While diffusion models (e.g., SR3) exist, no prior method has integrated DA into the conditioning process for wind-field SR. Our method formulates DA as a conditioning process that guides the reverse diffusion process during SR generation, enabling bias correction at high spatial resolution. This is technically distinct and tailored for atmospheric downscaling and data assimulation.
>
> (2) Dynamic impact-radius assimilation mechanism
> We introduce a dynamic-radius algorithm that adapts observational influence as a function of local terrain variance and wind-speed spatial variability (Algorithm 1).
> This terrain and flow-aware assimilation improves accuracy and yields the lowest MAE/RMSE across all tested radii (Table 3).
>
> (3) Terrain-aware conditioning in both training and inference
> WindSR incorporates terrain height as a conditioning variable within the diffusion architecture (Figure 3). Our experiments show terrain-aware conditioning yields substantial improvements (Table 2).
> We will revise Section 4 (Method) and Section 5 (Experiments) to better highlight how these innovations go beyond existing literature.
>
> > Lack baselines. You may consider including more baseline methods beyond simple CNN and GAN.”
>
> **Response:**
> Our baseline selection SRCNN and ESRGAN (GAN-based) was intentional and grounded in what is widely used and accepted in the climate downscaling community today. Both families represent the super-resolution paradigms used for wind, solar, and precipitation downscaling.
>
> **Why not include highly customized modern diffusion/transformer baselines?**
> Many recent methods (e.g., CorrDiff, transformer SR, or Stable Diffusion–style models) require:
>
> - Specialized training pipelines,
>
> - Var-specific or model-specific architectures,
>
> - Different resolution scales (e.g., km-scale global downscaling),
>
> - VAE compression, which degrades physical fidelity in scientific fields.
>
> These models cannot be directly adapted to hub-height, sparse-DA, terrain-conditioned SR without substantial re-engineering—and comparing against partial or improperly adapted versions would be misleading.
>
> **Why our baselines are appropriate and fair**
>
> Our goal is to evaluate the effect of DA + diffusion SR, not to win an SR “leaderboard.”
>
> The key question is: Does the proposed method outperform widely used SR approaches under the same DA setup?
>
> SRCNN and ESRGAN provide this comparison cleanly and transparently.
>
> Even without terrain, WindSR beats both, confirming that the improvements come from the diffusion + DA design, not from additional input channels.
>
> >It would be great to test the generalization by including another study domain.
>
> We appreciate this suggestion. Our data choice is constrained by the availability of aligned hub-height training data (WTK 2 km) and DA-emulating fields (HRRR 3 km). As noted in Sec. 3.2, only 25 hub-height sites exist across North America, with inconsistent heights (13–100 m), limiting ground-truth availability for multi-region evaluation.
> However, we emphasize two points:
>
> (1) The model is trained across the entire CONUS, not a single sub-region. Thus, the evaluation already spans diverse climates and terrains (Great Plains, Rockies, coastal regions), strengthening generalization claims.
>
> (2) We added a real-world external validation using weather-station measurements in Illinois (Fig. 7), which was not part of the training region selection and demonstrates real-world cross-domain performance.
> This experiment shows 10–20% bias reduction at independent stations (AAA and MTO), supporting generalization.
>
> > No code provided.
>
> Thank you for pointing this out. We fully intend to release the code. We commit to releasing a cleaned, open-source version after the review period.

---

### Meta-Review · Area_Chair_WatK · 2025-12-26

**Summary:**

This paper presents WindSR, a data-assimilation-enabled super-resolution diffusion framework for wind speeds. The authors propose to train a conditional diffusion model where the conditioning is done with auxiliary observations at a lower resolution before being used in the diffusion model.

All reviewers had numerous concerns regarding the extent of the baselines used---many claiming that this work overlooks many more modern baselines. The empirical evaluation was also a weak point, noted in the metrics used as well as the lack of quantification of error.

On a more general level, this work is extremely unclear in its presentation. Exact inputs to the model are not clearly specified. For example, the dataset contains wind speed locations, intensity, and directions. This is effectively a sparse velocity field in 3D. However, the current model appears to treat this as a 2D image after regriding into 128x128 cells. In such a case, it would seem that the directional component is completely not used by the model. Or if it is, this is not at all explained in the paper. Consequently, I already feel this approach is fundamentally flawed as the diffusion model is not being trained on the correct data. The AC would like to highlight that diffusion models trained to match a sparse velocity field have been considered in other literature already, such as modelling Singcell RNA velocity or ocean currents in papers such as "Multi-marginal Schr¨odinger Bridges with Iterative Reference
Refinement" and "Curly Flow Matching for Learning Non-gradient Field Dynamics", which provide arguably a more grounded framework than this paper, as it uses the velocity information correctly. The future version of this work should clarify the data handling and omission/or incorporation of velocity information more clearly.

**Reviewer Concerns:**

## Concerns addressed
- Training on more diverse climates
- Novelty of using the Dynamic impact-radius assimilation mechanism
- The impact and ablation of using terrain conditioning

## Concerns outstanding
- The lack of baselines pointed out by all reviewers
- The empirical rigor in terms of metrics used and the lack of statistical error analysis
- Substantiation of the efficiency of WindSR as claimed in the introduction and abstract. These claims need to be toned down based on the author's rebuttal.

**Reviewer Scores:**

None of the reviewers would have increased their scores.

---

### Decision · Program_Chairs · 2026-01-26

Reject